# Acceptability of decentralizing childhood tuberculosis diagnosis in low-income countries with high tuberculosis incidence: Experiences and perceptions from health care workers in Sub-Saharan Africa and South-East Asia

**Basant Joshi**[1]*, **Yara Voss De Lima**[2], **Douglas Mbang Massom**[3], **Sanary Kaing**[4], **Marie-France Banga**[5], **Egerton Tamba Kamara**[6], **Sheriff Sesay**[6], **Laurence Borand**[4¤], **Jean-Voisin Taguebue**[7], **Raoul Moh**[5], **Celso Khosa**[2], **Guillaume Breton**[6], **Juliet Mwanga-Amumpaire**[8], **Maryline Bonnet**[9], **Eric Wobudeya**[10], **Olivier Marcy**[1], **Joanna Orne-Gliemann**[1]*, on behalf of the TB-Speed Decentralization study group[¶]

**1** National Institute for Health and Medical Research (INSERM), Research Institute for Sustainable Development (IRD), University of Bordeaux, Bordeaux Population Health Centre, Bordeaux, France, **2** Instituto Nacional de Saúde, Marracuene, Mozambique, **3** IRD, Yaoundé, Cameroon, **4** Institut Pasteur du Cambodge, Epidemiology and Public Health Unit, Phnom Penh, Cambodia, **5** Programme PAC-CI, Site de l'ANRS Abidjan, Abidjan, Côte d'Ivoire, **6** Solthis, Freetown, Sierra Leone, **7** Fondation Chantal Biya, Centre Mère-Enfant, Yaoundé, Cameroon, **8** Epicentre Mbarara Research Centre, Mbarara, Uganda, **9** University of Montpellier, IRD,–INSERM, TRANSVIH MI, Montpellier, France, **10** MUJHU Research Collaboration, MU-JHU Care Limited, Kampala, Uganda

¤ Current address: Center for Tuberculosis Research, Division of Infectious Diseases, Johns Hopkins University, School of Medicine, Baltimore, MD, United States of America
¶ TB-Speed Decentralization study group members are listed in the acknowledgments section
* basant.joshi@u-bordeaux.fr (BJ); Joanna.orne-gliemann@u-bordeaux.fr (JO-G)

## Abstract

Decentralizing childhood tuberculosis services, including diagnosis, is now recommended by the WHO and could contribute to increasing tuberculosis detection in high burden countries. However, implementing microbiological tests and clinical evaluation could be challenging for health care workers (HCWs) in Primary Health Centers (PHCs) and even District Hospitals (DHs). We sought to assess the acceptability of decentralizing a comprehensive childhood tuberculosis diagnosis package from HCWs' perspective. We conducted implementation research nested within the TB-Speed Decentralization study. HCWs from two health districts of Cambodia, Cameroon, Côte d'Ivoire, Mozambique, Sierra Leone, and Uganda implemented systematic screening, nasopharyngeal aspirates (NPA) and stool sample collection with molecular testing, clinical evaluation and chest X-ray (CXR) interpretation. We investigated their experiences and perceptions in delivering the diagnostic package components in 2020–21 using individual semi-structured interviews. We conducted thematic analysis, supported by the Theoretical Framework of Acceptability. HCWs (n = 130, 55% female, median age 36 years, 53% nurses, 72% PHC-based) perceived that systematic screening, although increasing workload, was beneficial as it improved childhood

**Data Availability Statement:** The TB-Speed project is sponsored by the Inserm, the French National Institute of Health and Medical Research. We are therefore obliged to abide by French and European legal restrictions such as French law no. 78-17 of January 6, 1978 relating to data processing, files and freedoms usualy called "Loi informatique et libertés" and European law "GDPR" General Data Protection Regulation that govern data sharing and stipulate that data cannot be shared in open access to an undeclared thrid party if study participants have not been informed and did not give their consent for it. For the TB-Speed project, consent forms signed by participants indicated that " The written and/or recorded data that you will provide will be transferred securely to a database located in France for storage and analysis. This data will be handled according to the European and French regulation for the protection of personal information. Only authorized medical and study staff will have access to the database." Thus, persons wishing to have access to the interviews data may submit a request to the study investigator (Joanna.Orne-Gliemann@u-bordeaux.fr) and the sponsor (benjamin.hamze@inserm.fr), who will take steps to assess whether the request can be fulfilled. That said, we would like to emphasize that all qualitative data allowing to answer our study questions are presented in the paper.

**Funding:** The TB-Speed Decentralization Study is funded by UNITAID (2017-15-UBx-TB-Speed). The funders and sponsor did not play any role in the study design; data collection, data management, data analysis, data interpretation; report writing or the decision to submit reports for publication.

**Competing interests:** The authors have declared that no competing interests exist.

tuberculosis awareness. Most HCWs shared satisfaction and confidence in performing NPA, despite procedure duration, need to involve parents/colleagues and discomfort for children. HCWs shared positive attitudes towards stool sample-collection but were frustrated by delayed stool collection associated with cultural practices, transport and distance challenges. Molecular testing, conducted by nurses or laboratory technicians, was perceived as providing quality results, contributing to diagnosis. Clinical evaluation and diagnosis raised self-efficacy issues and need for continuous training and clinical mentoring. HCWs valued CXR, however complained that technical and logistical problems limited access to digital reports. Referral from PHC to DH was experienced as burdensome. HCWs at DH and PHC-levels perceived and experienced decentralized childhood tuberculosis diagnosis as acceptable. Implementation however could be hampered by feasibility issues, and calls for innovative referral mechanisms for patients, samples and CXR.

## Introduction

Childhood tuberculosis is a major global public health problem. Of an estimated 1.2 million children with tuberculosis in 2021, only around 58% were notified to the World Health Organization (WHO) [1]. This reporting gap is largely due to massive under-diagnosis that fuels high mortality in children [2,3]. The diagnosis of tuberculosis in children is challenging due to several individual, biomedical, and structural reasons: 1) despite being possibly severe, tuberculosis is often paucibacillary in children, hence poorly detected, even by recent molecular tests [4,5], and respiratory samples for diagnosis are challenging to collect as young children are unable to self-expectorate sputum thus require more complex sample collection methods such as gastric aspirates [6]; 2) there is no point-of-care test for childhood tuberculosis to tackle the microbiological diagnosis gap; 3) signs and symptoms used for clinical diagnosis of tuberculosis have poor sensitivity and specificity in children, and good quality chest X-ray and reliable reading is poorly accessible; 4) lastly, equipment and expertise needed for the diagnosis of childhood tuberculosis is usually lacking at lower levels of health care where most children access services [7].

WHO recently recommended decentralized models of care to deliver tuberculosis services to children and adolescents with signs and symptoms of tuberculosis [8,9]. This recommendation was based on a systematic review of studies showing that strengthening both diagnostic capacity at low levels of care, and linkage between communities and health facilities, had a positive impact on the notification of tuberculosis [10–13].

As per the definition proposed by Mills et.al, the decentralization of health services means transferring the mandate and the capacity to diagnose tuberculosis from higher to lower levels of health facilities [14]. A number of tuberculosis diagnostic tools and interventions, either already recommended by WHO or being evaluated, are currently concentrated at tertiary care levels but could be decentralized at district hospital (DH) and primary health center (PHC) levels. Nasopharyngeal aspirates and stool samples could be suitable to peripheral health care levels, as good alternatives to gastric aspirates that usually require hospitalization [15]. Portable and battery-operated molecular testing devices (using Xpert MTB-RIF Ultra) are expanding at DH-level in most countries and can be used routinely at PHC-level [8]. Access to quality chest X-Ray and interpretation, one of the challenges at PHC-level and sometimes also at DH-level, could be improved by the use of digital systems to ease the transfer of images for interpretation by trained clinicians [16,17]. As HCWs at low level of health care may be overburdened, with sometimes limited clinical skills and capacity for diagnosis, decentralized models of care

require capacity-building and strengthening of clinical and treatment decision-making skills, which can be achieved by training, clinical mentoring and support supervision [18].

WHO called for more data on the feasibility and acceptability of decentralized models of care in order to guide implementers and decision-makers [9]. Acceptability is a key implementation outcome, capturing the perceptions among specific stakeholders of whether and how a given service or practice is agreeable or satisfactory [19–21]. Evaluating acceptability provides insights as to barriers and facilitators of different services and interventions, which can be useful for implementers and decision makers. For instance, in South Africa, acceptability findings on a tuberculosis literacy and counselling intervention provided insight for designing a task shifting model for patient-centered interventions aimed at improving retention within the cascade of tuberculosis care [22]. A multicenter study exploring the acceptability of a novel fixed-dose combination anti-tuberculosis drug reported on the good acceptance of this combination drug, but highlighted administration challenges, underlying the need to include structured initial and ongoing education and adherence support to caregivers, in future scale-up plans [23].

In the field of the diagnosis of childhood tuberculosis, many implementation questions remain unanswered. How do PHC-based nurses or clinicians engage in nasopharyngeal aspiration and stool sample collection, or in chest X-ray reading and clinical diagnosis? Do perceptions of HCWs differ whether they are based at DH or PHC levels, in one country or the other, and how does context influence their acceptability of new services? In order to support the implementation of decentralized childhood tuberculosis services, we sought to assess the acceptability of decentralizing a comprehensive childhood tuberculosis diagnosis package among HCWs at low levels of care.

## Materials and methods

### Study design and setting

As part of the TB-Speed Decentralization study, we conducted a multi-country and mixed methods, pre/post implementation research study. We report here on the post-intervention qualitative assessment.

The TB Speed Decentralization study aimed at assessing the impact on tuberculosis case detection of decentralizing a comprehensive childhood tuberculosis diagnosis package at DH and PHC-levels in countries with high tuberculosis incidence [24]. In Uganda and Mozambique, tuberculosis diagnosis services were partly decentralized at DH and even at PHC-level; this consisted mainly in smear microscopy performed on expectorated sputum for microbiological diagnosis and did not include NPA and stool sample collection and sample testing using Xpert Ultra. It was a before-after study, conducted between October 2019 and August 2022, in two rural or semi-urban health districts of six countries, namely Cambodia, Cameroon, Côte d'Ivoire, Mozambique, Sierra Leone and Uganda, including 59 health facilities (12 DHs and 47 PHCs). Prior to the start of the TB-Speed decentralization study, we conducted a baseline assessment survey in the six study countries to select districts and health facilities. Of the 179 PHCs surveyed in 32 districts from 5 countries (Cambodia, Cameroon, Cote d'Ivoire, Mozambique and Uganda), 75% of PHCs had specific staff members for TB management and only 30% were trained for pediatric TB. Although around 70% of PHCs collected microbiological sample on site, only 7% of them collected samples from children. Around 46% of PHCs had onsite laboratory with 3.3% having X-pert MTB/RIF in the laboratory. From this survey, districts for the study were selected with National TB Programs (NTPs) with the following criteria: 1) rural or semi-urban, 2) one DH and 4 PHC at least that could participate in the study; 3) have functional X-ray services or capacity to perform X-ray at DH; 4) minimum human resource of a clinician, a nurse and a laboratory technician at PHC level; 5) not receiving

support from NGOs or other institutions except if in line with NTP guidelines; this selection also took in consideration the tuberculosis case notification and population size of districts. Overall 12 DHs and 47 PHCs were selected including, 34% that were tuberculosis diagnosis and treatment units. Two levels of decentralization were implemented in each country, i.e. a DH-focused and a PHC-focused strategy, based on the hypothesis that decentralizing tuberculosis diagnosis down to the PHC level would improve access but could face additional feasibility challenges compared to DH, related to staff availability and capacity. The study was therefore designed to formally assess and compare the effect on tuberculosis case detection and implementation challenges of the two proposed strategies. In DH-focused districts, the DH and 4 PHCs conducted systematic symptom-based screening for tuberculosis among children, using simple questions asked either at triage and/or in consultation rooms. Children identified as presumptive tuberculosis at PHC-level were referred to DH (parents did not receive transport reimbursement) for Xpert MTB-RIF Ultra (Ultra) testing on nasopharyngeal aspirate, stool and/or expectorated sputum samples, as well as strengthened clinical evaluation, digital chest X-Ray with a simplified reading approach, and final diagnosis of tuberculosis. Details of the activities conducted by HCWs are presented in Table 1. In PHC-focused districts, the DH and 4 PHCs were equipped to implement the comprehensive childhood tuberculosis diagnosis package; only chest X-Ray was done at DH-level for all children seen at DH and in those that were still symptomatic after a week at PHC (parents received transport reimbursement); images were made available to clinicians in PHCs. Interventions were implemented based on an international and country specific implementation guides with local adaptations.

## Study population

A purposive sample of 1 to 5 HCWs from each participating health facility was recruited for interview. HCWs were selected considering the total staff number in each facility, and based on their experience in delivering components of the childhood TB diagnosis package within their health facility (screening, clinical evaluation, sample collection, sample testing, performing or reading chest X-Ray). Participants targeted were medical doctors, nurses, clinicians/paramedics, radiographers, laboratory technicians, and community health workers. SSRAs were advised to recruit additional staff if they believed it would provide interesting insight and additional quality data (i.e. purposively recruiting laboratory technicians or radiographers in the facilities that had such staff).

## Data collection

Trained Social science research assistants (SSRAs) in each country liaised with the health facility in-charge or manager, to contact eligible HCWs and sought their written informed consent. Sociodemographic information was collected from all participating HCWs. Interviews were conducted face to face or by phone (when transport to site was impossible due to COVID-19 restrictions) and were recorded if respondents permitted. Interviews were conducted using a semi-structured guide exploring perceptions and experience of HCWs in implementing each component of the comprehensive childhood tuberculosis diagnosis package in their health facility (S1 File). We piloted the interview guide in one country (Sierra Leone), during two interviews, and discussed the process within the multi-country team of SSRAs before finalizing a standard version of the guide, to be used in all six countries. Interview audios, transcripts and translated transcripts (if the interview was conducted in a language other than English) were uploaded on a secured FTPS server at University of Bordeaux, France. Interviews were completed in a single session, though they were sometimes interrupted by patients or other HCWs requiring the

**Table 1. Comprehensive childhood tuberculosis diagnosis package.** TB-Speed Decentralization study (2018–22).

| Diagnosis component | Target population | Person in charge | Location | Detail of activity | DH-focused strategy | | PHC-focused strategy | |
|---|---|---|---|---|---|---|---|---|
| | | | | | DH | PHC | DH | PHC |
| **Systematic Screening** | All sick children attending the health facilities | Screener (trained lay health care), Nurse, Nurse assistant | Triage or consultation room | Screening questions:<br>1. Cough > 2 weeks<br>2. Fever > 2 weeks<br>3. Documented/reported weight loss<br>4. Contact with a person with tuberculosis and any cough<br>Facility or study specific screening registers | X | X | X | X |
| **NPA sample collection** | Children with presumptive tuberculosis | Nurse, Nurse assistant, Midwife | Sample collection room or consultation room | Microbiological samples: nasopharyngeal aspirate, expectorated sputum (if stool is not available) | X | | X | X |
| **Stool sample collection** | Children with presumptive tuberculosis | Nurse, Nurse assistant, Midwife | Sample collection room or consultation room | Stool sampling (directly from the child or with the help of parents on the day or later) | X | | X | X |
| **Microbiological testing using Xpert Ultra** | Children with presumptive tuberculosis | Laboratory technician (DH) Nurses/midwives or laboratory technician (PHC) | Laboratory, NPA collection room (in some PHCs) | Xpert MTB/RIF Ultra on:<br>• Battery-operated G1 Edge (PHC)<br>• Standard GeneXpert (DH) | X | | X | X |
| **Clinical evaluation** | Children with presumptive tuberculosis | Clinician, nurse, medical doctor | Consultation room | Standardized medical charts with symptoms and signs of tuberculosis.<br>In Uganda NTP algorithm for the diagnosis of tuberculosis | X | | X | X |
| **Optimized chest X-Ray** | All children with presumptive tuberculosis (DH) Still symptomatic at Day 7 (PHC) | Radiographer (Performs CXR), Team (Medical doctor, clinician, Nurse) led by Medical doctor (Interpret CXR report) | Radiology department, consultation room | Digital chest X-Ray & simplified reading<br>1. Enlarged lymph nodes<br>2. Alveolar opacity<br>3. Airways compression<br>4. Miliary<br>5. Cavitation<br>Pleural or pericardial effusion | X | | X | |

DH: District Hospital, PHC: Primary Health Clinic.

respondent's input. The duration of interviews usually ranged between 1 hour and 1 hour and 30 minutes. Study forms and codes were protected and only handled by study researchers. Multi-country data collection coordination meetings were conducted between SSRAs, the implementation research coordinator and the implementation research investigator after the first two interviews in each country, and on a regular basis to maximize the quality of interviews conducted.

## Ethical aspects

This study is part of clinical study C18-25 sponsored by Inserm (Institut National de la Santé et de la Recherche Médicale, Paris, France). It was granted approval by the Inserm Institutional Review Board (IRB00003888) and WHO ethics advisory board at international level and each country ethics committees and registered in a public study registry (NCT04038632). All study participants gave their informed, written consent to participation, in line with ethical guidelines. Study protocol and tools were approved by Inserm Institutional Review Board and WHO ethics advisory board at international level and each country ethics committees.

### Data analysis

We conducted thematic analysis of the interviews. Deductive coding was conducted using a codebook derived from a) the interview guide topics–including each key component of the comprehensive childhood tuberculosis diagnosis package; and b) the Sekhon et al. Theoretical Framework of Acceptability [20]–this framework proposes seven key acceptability domains: affective attitude, burden, ethicality, intervention coherence, perceived effectiveness, opportunity costs and self-efficacy. Data were coded within the NVivo Release qualitative data management software, QSR International Private Ltd. Version 1.5, 2021. Multi-country intercoder reliability was maximized by SSRAs coding the same three transcripts and reviewing together the coded data. Findings were grouped into the deductive overarching thematic categories (a bi-dimensional matrix comprising the acceptability domains and the different tuberculosis diagnosis components), and summarized. Summaries were discussed with the country Principal Investigators, Program Managers and the International study coordinating investigators for their feedback and validation. During the interpretation process, findings were further categorized according to the levels of the Socio-Ecological Model [25,26].

## Results

Individual interviews were conducted with 130 HCWs between June and August 2021. The median age of respondents was 36 years (IQR; 31, 41) and 71 (55%) were female (Table 2). HCWs were mostly from PHCs (72%). Globally, more than half of the HCWs interviewed were nurses (ranging from 31% in Uganda to 87% in Côte d'Ivoire) followed by clinical officers/medical assistants (18%) and medical doctors (10%). The highest number of medical doctors interviewed were from Cameroon (21%) and none of the medical doctors were from Sierra Leone. Most HCWs reported having conducted systematic screening for tuberculosis (77%), and half experienced conducting clinical evaluation.

We describe below acceptability findings per component of the comprehensive childhood tuberculosis diagnosis package. Table 3 presents emblematic quotes, organized according to the different acceptability domains proposed by the Sekhon framework.

### Systematic screening

Almost all HCWs perceived systematic screening as a positive intervention contributing to improve awareness of childhood tuberculosis among HCWs and parents in health facilities.

*"It comes to arouse our attention to what is the screening of tuberculosis in children because this has been an Achille's heel for us. The nature of pediatric tuberculosis itself is very difficult to diagnose*

*(Mozambique, DH, Nurse).*

HCWs appreciated that systematic screening allowed them to be proactive about finding children with presumptive tuberculosis.

*"Well, it must be said that systematic screening has made it possible to identify cases because before, for example, we had to wait for a subject who necessarily presented us with a cough, but with the screening of tuberculosis we saw that the other entry points, the other signs and symptoms that could also be used, and in particular weight loss in children*

*(Côte d'Ivoire, DH, Doctor)*

**Table 2. Sociodemographic information of HCWs interviewed, TB-Speed Decentralization study, 2021.**

| Variables | Cambodia (N = 20) n (%) | Cameroon (N = 24) n (%) | Côte d'Ivoire (N = 16) n (%) | Mozambique (N = 19) N (%) | Sierra Leone (N = 22) n (%) | Uganda (N = 29) n (%) | Total (N = 130) n (%) |
|---|---|---|---|---|---|---|---|
| Gender | | | | | | | |
| Male | 10 (50) | 10 (42) | 14 (88) | 5 (26) | 6 (27) | 14 (48) | 59 (45) |
| Female | 10 (50) | 14 (58) | 2 (12) | 14 (74) | 16 (73) | 15 (52) | 71 (55) |
| Age (Years)- Median (IQR) | 33 (29–43) | 33 (30–41) | 37 (34–43) | 33 (28–38) | 37 (32–42) | 37 (33–40) | 36 (31–41) |
| Type of decentralization | | | | | | | |
| DH-focused | 10 (50) | 13 (54) | 9 (56) | 10 (53) | 12 (55) | 14 (48) | 68 (52) |
| PHC-focused | 10 (50) | 11 (46) | 7 (44) | 9 (47) | 10 (45) | 15 (52) | 62 (48) |
| Type of health facility | | | | | | | |
| District hospital | 7 (35) | 6 (25) | 3 (19) | 5 (26) | 6 (27) | 10 (34) | 37 (28) |
| Primary Health Clinic | 13 (65) | 18 (75) | 13 (81) | 14 (74) | 16 (73) | 19 (66) | 93 (72) |
| Position | | | | | | | |
| Medical doctor | 3 (15) | 5 (21) | 2 (13) | 1 (6) | 0 (0) | 2 (7) | 13 (10) |
| Clinician/Paramedics | 0 (0) | 0 (0) | 0 (0) | 9 (47) | 9 (41) | 5 (17) | 23 (18) |
| Nurse/Nursing assistant | 8 (40) | 19 (79) | 14 (87) | 9 (47) | 10 (45) | 9 (31) | 69 (53) |
| Midwife | 6 (30) | 0 (0) | 0 (0) | 0 (0) | 1 (5) | 0 (0) | 7 (5) |
| Laboratory technologist | 1 (5) | 0 (0) | 0 (0) | 0 (0) | 2 (9) | 6 (21) | 9 (6) |
| Radiographer | 0 (0) | 0 (0) | 0 (0) | 0 (0) | 0 (0) | 2 (7) | 2 (2) |
| Screener | 0 (0) | 0 (0) | 0 (0) | 0 (0) | 0 (0) | 5 (17) | 5 (4) |
| Other | 2 (10) | 0 (0) | 0 (0) | 0 (0) | 0 (0) | 0 (0) | 2 (2) |
| Experience in the health facility | | | | | | | |
| < 1 year | 0 (0) | 2 (8) | 4 (25) | 0 (0) | 0 (0) | 2 (7) | 8 (6) |
| 1 to 5 years | 8 (40) | 15 (63) | 5 (31) | 8 (42) | 16 (73) | 20 (69) | 72 (55) |
| 5 to 10 years | 6 (30) | 2 (8) | 4 (25) | 6 (32) | 4 (18) | 0 (0) | 22 (17) |
| > 10 years | 6 (30) | 5 (21) | 3 (19) | 5 (26) | 2 (9) | 7 (24) | 28 (22) |
| Involvement in intervention component | | | | | | | |
| Systematic screening | 14 (70) | 22 (92) | 16 (100) | 12 (63) | 20 (91) | 16 (55) | 100 (77) |
| Nasopharyngeal aspiration | 9 (45) | 13 (54) | 9 (56) | 5 (26) | 11 (50) | 11 (38) | 58 (45) |
| Stool sample collection | 11 (55) | 15 (63) | 6 (38) | 5 (26) | 12 (55) | 7 (24) | 56 (43) |
| Microbiological testing (Xpert MTB-RIF Ultra) | 8 (40) | 5 (21) | 8 (50) | 10 (53) | 7 (32) | 8 (28) | 46 (35) |
| Chest X-Ray | 12 (60) | 5 (21) | 8 (50) | 10 (53) | 5 (23) | 14 (48) | 54 (42) |
| Clinical evaluation | 7 (35) | 10 (42) | 4 (25) | 15 (79) | 21 (95) | 12 (41) | 69 (53) |

Although the majority of HCWs were confident in performing systematic screening, some reported facing poor understanding, and sometimes reluctance from parents or accompanying persons (older children or grandparents) when responding to screening questions on weight loss, contact history, or days with fever. HCWs, especially those working at PHCs, found screening documentation burdensome in the context of busy schedules and competing priorities with insufficient staff.

*"The documents we fill out are too much. You do the screening and documents to fill out are many, so I find it difficult to remember, another staff fills them out. If we are two staff, we do it a whole morning. (. . .) Two staffs who explain and fill out many documents. It is not useless, but it is just waste of the time".*

*(Cambodia, DH-focused district, PHC, Nurse)*

**Table 3. Acceptability of decentralizing a comprehensive childhood tuberculosis diagnosis package in six low-income and high burden countries (2017–2022).** Quotes organised according to the Sekhon Theoretical Framework of Acceptability.

| Acceptability domain | Diagnosis component | Illustrative quotes |
|---|---|---|
| **Affective attitude** | **Screening** | *"The feelings? Well, they are feelings of joy, it (screening) helped us and we really took it with joy because (…) if this project was not there, we were really going to let children go, perhaps we were going to treat them for other pathologies whereas it was tuberculosis that was tiring them" (Côte d'Ivoire, PHC-focused district, PHC, Nurse).* |
| | **Referral** | *"Most of the parents that have gone through the procedure appreciate it, and now they are the ones telling another parent who cries when referred to the government hospital that 'look at my baby now he is very healthy, and you are not going to be admitted at the hospital, so please listen to (name of health worker), and the man that they will send you to will not delay you as it uses to be, and that man will do it faster for the children" (Sierra Leone, DH-focused district, PHC, Clinical Officer).* <br> *"But we as health professionals at some point we are obligated to insist but not in a rude way. Make them understand that we are referring because we lack some resources and that there because it is a bigger hospital, they will get better". (Mozambique, DH focused district, PHC, Nurse).* |
| **Perceived-effectiveness** | **Screening** | *"I'm telling the truth, it's true that at our level we couldn't imagine that little children could have tuberculosis. To think that a child with a cough could have TB… With the project, we knew that even children could suffer from it, so when (during screening) we met a child who met the criteria, it required a referral for a probable search for TB" (Cameroon, DH-focused district, PHC, Nurse)* |
| | **Nasopharyngeal aspiration** | *"It was difficult to diagnose, for example, the children who could not manage to give sputum, it (nasopharyngeal aspiration) helped a lot. I may not be able to explain it in my own words, but overall it is helping, it is improving, it is making the health center grow." (Mozambique, PHC-focused district, PHC, General medical technician)* <br> *"Yes, it (nasopharyngeal aspiration) contributes to identify tuberculosis in children, it does help a lot. Because we aspirate the sample from a child and test for tuberculosis. If the child is positive then we can provide the treatment." (Cambodia, PHC focused district, PHC, Midwife)* |
| | **Chest X-Ray** | *So chest X-Ray has reduced the mortality rate first of all, improved the health of these children and improved the perception among mothers, that tuberculosis is there and it can be with their children regardless of the parent being with tuberculosis (or not). Because they don't know the contacts where they got it from but you find them when they are positive, they get treatment and they become fine." (Uganda, DH, Nurse).* <br> *"Yes, the X-ray helped because there were some children who could not find the germ in the stool or in the nasopharyngeal aspiration but it was the X-ray that showed their positivity." (Cameroon, DH, Nurse)* |
| **Burden** | **Nasopharyngeal aspiration** | *"For NPA, I've already done it once and I can say that it's an unpleasant thing, honestly. It's unpleasant to have to hold a child, send the tubes through the nostrils and do it in front of the mother (…) I just felt pity for the child but I had to do the exam, I had to remain professional and it was also for his sake." (Cameroon, PHC-focused district, PHC, Nurse)* |
| | **Referral** | *"When we have to refer them to the general hospital. Some parents, when you explain to them, well they say (…) my husband is not here so they can't go." (Côte d'Ivoire, DH-focused district, PHC, Nursing assistant)* <br> *"There are (some) who are referred but are not able to reach the central health facility. Because they do not have the amount (of money) to be able to catch public transport. There are health units that are really very far from the central health facility. The inhabitants here are people who are very disadvantaged." (Mozambique, DH, Nurse).* |
| **Self-efficacy** | **Xpert MTB-RIF Ultra** | *"I never do it as I am afraid of breaking the machine (G1 Edge). They told me to see and follow the steps but I am still afraid to spoiling the machine. I don't dare to do it." (Cambodia, PHC-focused district, PHC, Midwife)* |
| | **Chest X-Ray** | *"So, there's training that we've received and with the clinical mentorship we've really been able to improve, because I myself feel that, I'm a little more comfortable reading a pediatric X-ray now than I was a while ago." (Cameroon, DH, General Practitioner).* |
| | **Clinical evaluation** | *"I feel more confident. I give it much time. After the training, I got more knowledge to go much deeper while evaluating presumptive children to find out whether the child is having TB or not." (Uganda, DH-focused district, PHC, clinical officer)* |
| **Ethicality** | **Nasopharyngeal aspiration** | *"It (nasopharyngeal aspiration) is less invasive, (compared to) the gastric aspirate that we used to get before. Because personally I used to use the gastric to collect samples for tuberculosis detection in children." (Uganda, DH, General Practitioner).* <br> *"Apart from the fact that sometimes after taking a sample from the first nostril, the child starts to manifest or he feels uncomfortable, and at times we have to stop. These are the only difficulties we have encountered." (Cameroon, PHC-focused district, PHC, Nurse)* <br> *"So far no. Nobody has refused. (…) now it is a study, but in the ward, it wouldn't be a bad thing if there was another nurse who could learn how to do it. Because people see it as a very particular technique that is very special". (Mozambique, DH, Clinician)* |

*(Continued)*

**Table 3.** (Continued)

| Acceptability domain | Diagnosis component | Illustrative quotes |
|---|---|---|
| Intervention coherence | Nasopharyngeal aspiration | *"So, you have to convince the parent first before doing the nasopharyngeal aspiration, and also always measure the saturation before, the parameters, it's important to do it before the aspiration. Because if you do a sample and then there are problems and they ask you what the patient's parameters were and you can't give them, it means that you yourself have contributed to this child going down."* (Cameroon, DH, Nurse) |
| | Chest X-Ray | *"Because now even if someone is having a negative result, and we look at the x-ray, we can still know that this is tuberculosis or not. Especially those whom we give antibiotics and they still fail and they come we repeat a chest X-ray and we find some opacities in the lungs then we treat them clinically for tuberculosis."* (Uganda, DH, Nurse) |
| Opportunity cost | Screening | *"The children who have fallen under the process I still follow them up to the village. I ride to their homes, I inspect the children then if I see it is required for that very person, that child to be brought back."* (Uganda, DH-focused district, PHC, Screener). |

### Nasopharyngeal aspirate collection

Nasopharyngeal aspiration was perceived as an easy and innovative point-of-care respiratory sample collection method in children unable to expectorate sputum. HCWs found that nasopharyngeal aspiration was less invasive than other procedures.

*"It (nasopharyngeal aspiration) has helped in the sense that there are some children who cannot produce the sputum and you can only go through that method. Before I know they used to do the gastric tube and I think that's more painful than the nasopharyngeal aspiration"*

*(Cameroon, DH, Nurse)*

Most HCWs, even at DH-level, did not have prior experience of nasopharyngeal aspiration. Some faced challenges with the procedure (collecting bloody samples for example), especially during initial days. But HCWs felt that training and practicum session had contributed to increase their skills and overall shared their satisfaction and confidence in performing nasopharyngeal aspiration.

*"Like for me, previously I told you that I was afraid to collect the NPA (nasopharyngeal aspirate) doing it through the nostril, you know, nostril. Fortunately for me, (name of country project manager) was here and I had two kids in front of them (project team), and I did one while they also did the other one. I learned from them"*

*(Sierra Leone, PHC-focused district, PHC, Community Health Officer)*

Some HCWs shared concern about their lack of practice of nasopharyngeal aspiration during the COVID-19 pandemic and the related interruption in health services.

HCWs perceived that performing nasopharyngeal aspiration required some effort, associated with the duration of the procedure, the need to involve parents/colleagues (for restraining the baby/child during the procedure), managing discomfort caused to children, or preventing bloody samples.

*"But then the challenge is when the child is fighting, you have to get more than three people to assist you, whereby they need also the gadgets (i.e. equipment). Because there those ones of 8 years, 9 years, 10 years, they are powerful even when they are sick so you have to get people to assist you on performing that nasopharyngeal aspiration."*

*(Uganda, DH, Nurse).*

However, almost all HCWs, whether from DHs or PHCs, perceived nasopharyngeal aspiration as quick and efficient, contributing to diagnosis and treatment of tuberculosis.

*"It's (nasopharyngeal aspiration) quick and you don't have to handle it like we often do with sputum and stuff. . . It's quick and they give you the results right away and if the child is sick, they put them on medication directly. We appreciated that"*

*(Cameroon, PHC-focused district, PHC, General Practitioner).*

## Stool sample collection

HCWs had positive attitudes towards stool sample collection. They perceived it as effective, with instances of positive tuberculosis results found on stool samples, whereas testing on nasopharyngeal aspirate had been negative. They acknowledged however that stool collection is mostly beyond their personal control as HCWs, and even that of parents, as children cannot produce stools on demand. They reported frustration towards parents delaying collection due to cultural practices, impossibility or unaffordability to travel long distances, or because of other priorities.

*"In our rural context, there are certain habits that can explain delays in getting stool samples; it is common practice for our mothers to give "evacuating enemas" to children. So, when the child arrives at the clinic he has already finished evacuating stools for the day. You have to wait until the next day and hope to be able to collect stool. Why are they doing this?"*

*(Côte d'Ivoire, DH, doctor)*

Some HCWs underlined improper handling of stool samples and containers by parents, which was perceived as impairing sample processing at DH laboratory.

*"They (parents) did not much pay attention, often they took it full container, dirty and bad smell like that."*

*(Cambodia, DH, Pediatric doctor)*

## Microbiological testing of samples with Xpert Ultra

Ultra was perceived by all HCWs as a high quality and effective test to improve diagnosis (including identification of rifampicin resistance) and increase early childhood TB diagnosis.

*"What I can say is that Xpert is a method (. . .) that is helping a lot because it already gives us a lot of assurance for us to know that in fact it is negative, we are sure it is. If it's positive, it's positive. The sensitivity that the Xpert Ultra has is something really good. It's something good because it showed us that, for example, for the stools that we had are positive. It showed us that it is possible to detect the bacillus in an easier way and with other types of samples besides the expectoration and nasopharyngeal aspirates"*

*(Mozambique, DH, General Medical Technician)*

The availability of childhood tuberculosis diagnosis through Ultra was reported as increasing the trust of parents towards HCWs and the quality of services provided at low levels of care.

*"The advantage is that children don't need to go to referral hospital. We can do it (Xpert) at the health facility. So, it promotes our health facility services. People are appreciating our service here, that we can provide many services. If they cannot seek services they want, our health facility will not be well spoken of, and we will have low service.*

*(Cambodia, PHC-focused district, PHC, Midwife)*

HCWs operating in the laboratories were sometimes reluctant to process stool samples (at DH-level) or to test poor quality nasopharyngeal aspirates (especially at PHC-level). When receiving large numbers of samples, or because of power cuts, they were faced with increased turnaround time, especially in PHCs equipped with one module Ultra machines.

*"The turnaround time is long because this machine runs about four samples a day, yet you find we have about fifteen samples to process in one day. In fact, it sometimes forces us to use ZN [Ziehl-Neelsen stain method for smear microscopy] instead of the GeneXpert machine for the sake of ensuring that other patients get their results in time."*

*(Uganda, PHC-focused district, PHC, Laboratory technician)*

## Referral of children and samples

Most HCWs working at PHCs of DH-focused districts shared frustration with parents either refusing referral, fearing referral (associated in their minds with the severity of a disease) or being unable to reach DH due to distance and transportation challenges, or other priorities of their daily life/work.

*"Those who come they appreciated the (health) system (...) but the only problem if we ask them to go for further management, they raise a problem. They say that for me we don't have money, we have other children to look after so if I go there who will look after these young ones so, now I don't have anybody, they bring excuses."*

*(Uganda, DH-focused district, PHC, Nurse).*

This was particularly exacerbated during the COVID-19 pandemic, as parents did not easily go to DH due to fear of COVID-19 (most DHs being COVID management centres) and inability to travel in periods of lockdown.

Referral to DH was experienced as burdensome for HCWs at PHC-levels, especially when the child was accompanied by grand-parents/siblings (consent for referral could not be obtained). HCWs explained that parents required intensive referral counselling and sometimes needed to be escorted to DH to ensure proper onwards management of their child.

*"Referral requires a lot of tact and time, and you have to flatter the patient so that he can go to DH. So, screening is easier than referral"*

*(Cameroon, DH-focused district, PHC, Nurse)*

*"At the general hospital there are crowds of people, so always we have them (parents) accompanied by one of our collaborators (colleague at PHC), to be able to locate them. Otherwise when you tell them to go there themselves, they will go and sit in the hospital forever. (...)*

*There are even difficulties, but I always take one of my collaborators, parents have to be supported"*

*(Côte d'Ivoire, PHC-focused district, PHC, Nurse)*

In response to these challenges, some PHC-based HCWs explained that they sometimes ended up not referring parents to the DH at all.

HCWs from PHCs in the DH-focused districts also reported challenges in sending collected stool samples to DH. In Uganda, HCWs reported adapting to those barriers by referring both samples and patients to DH at the same time (parents being directly in charge of sample transport). Delays in receiving test results from DH for referred children, poor communication with HCWs at DH, and with parents of referred children, were some of the challenges experienced post-referral by HCWs of PHCs.

## Clinical evaluation

HCWs shared positive attitudes towards clinical evaluation as a key step for diagnosing tuberculosis in a child. Clinical evaluation was perceived as complementary to laboratory and radiological tests, with each of these diagnostic "tools" playing a different role and being more or less prominent according to individual situations. Some HCWs reported conducting more thorough clinical evaluation in search of tuberculosis, following a "gut feeling" that microbiological testing had failed to detect tuberculosis.

*"Yes, I think it really helps because the evidence is that we had one case where we didn't have a positive result with nasopharyngeal aspiration and with stool, but it was because of the clinical evaluation that we were able to explain that to our doctor, to the point where we sent the child to do the x-ray and it came back positive"*

*(Cameroon, PHC-focused district, PHC, Nurse)*

HCWs reported collaborative discussions around the evaluation of children. Some HCWs from PHC level reported increased confidence in their clinical evaluation skills, following the decentralization of services and associated training and mentoring, being able to anticipate on a confirmatory final diagnosis from DH-level clinicians, regarding the children they referred.

*"Yes, I'm comforted by the fact that there are cases that have been sent to the district hospital with a high degree of certainty that they will be positive cases, based on clinical assessment. Because we investigate much more thoroughly and we know that the person is sick"*

*(Cameroon, DH-focused district, PHC, General Practitioner)*

## Chest X-Ray

The majority of HCWs valued chest X-Ray due to its contribution to the diagnosis of tuberculosis in children and improvement in their health status. They reported trusting chest X-Ray for decision-making and final diagnosis.

*"And because, you don't go into details because you are not a medical person, but we use the chest X-Ray to be our last resource to have our final diagnosis for the presumptive cases, so with that result now, when that result is out, it gives us the final diagnosis."*

*(Sierra Leone, DH, Nurse).*

For many HCWs, using chest X-Ray was a new clinical practice, and reading images was a new skill. HCWs appreciated the training, support and mentorship in increasing confidence and self-efficacy in performing and reading chest X-Rays.

*"After getting training on how to read chest X-Ray films, we thought it was a miracle. The first chest X-Ray result I read was negative. I am the one who read it and when the team from TB speed came for supervision, they said it was true and I was very happy. I have got some experience in reading chest X-Ray films although there are some challenges associated with it."*

*(Uganda, PHC-focused district, PHC, Nurse).*

However, HCWs from PHCs reported insufficient exposure in reading chest X-Rays as compared to DH-based HCWs. Some reported lack of feedback from DH about the child they referred for chest X-Ray, which resulted in lack of confidence and demotivation to refer children for chest X-Ray.

*"If we do it (CXR interpretation) more and more, I think, it is possible. We are skillful. But we do one and long time later do another one, I got some confusing. Yes, any work we do as routine, we know it well. But we do one, and a long time later we do another one, I get confused."*

*(Cambodia, PHC-focused district, PHC, Midwife)*

Referral to DH for chest X-Ray was perceived as challenging, despite of travel fees reimbursement provided to parents referred for chest X-Ray. Yet some HCWs perceived that establishing chest X-Ray at PHCs might not be adequate either.

*"Chest X-Ray can be done at the district hospital, so that the children we refer to the district hospital are able to access the service from that side (. . .). Because we receive few clients and it may not be cost effective to have the chest X-Ray machine at this facility".*

*(Uganda, DH-focused district, PHC, Clinical Officer).*

## Discussion

In this multi-country qualitative study, HCWs shared overall positive perceptions of decentralizing a comprehensive childhood tuberculosis diagnosis package, appreciating their increased knowledge and skills, valuing each diagnostic tool and its contribution to improving diagnosis. This confirms the pre-intervention findings on the a priori acceptability of decentralized diagnosis of childhood tuberculosis among HCWs [27]. However, the experience of HCWs shared during individual interviews was more nuanced, with challenges reported, feelings of frustration and of being faced with many external barriers in implementing childhood tuberculosis diagnosis at low levels of care. Our study results show that the overall acceptability of decentralizing childhood tuberculosis diagnosis is influenced by individual, health system and structural factors (Table 4).

At an individual level, the acceptability of the comprehensive childhood tuberculosis diagnosis package was driven by both emotional and rational/cognitive reasons. HCWs in all countries and all facilities shared feelings of hope that this comprehensive diagnosis package was effective and improving the detection of tuberculosis. Decentralizing the diagnosis of tuberculosis was perceived as ethical as services were brought closer to people, which meant increased

**Table 4. Facilitators and barriers to the acceptability of decentralizing a comprehensive childhood tuberculosis diagnosis package (2017–2022).**

| | Facilitators | Barriers |
|---|---|---|
| **Individual-level** | • Bringing services closer to people in need<br>• Self-efficacy in diagnosing tuberculosis<br>• Trusting the quality of sample collection procedures (NPA, stool collection) and the tests (Xpert Ultra)<br>• Being pro-active in detecting tuberculosis<br>• Observing tangible results, positive tests, effectiveness<br>• Feeling proud to provide quality services | • Causing discomfort to children during the nasopharyngeal aspiration procedure<br>• Lack of exposure to some new diagnostic tools and confidence<br>• Not being able to solve referral related issues (transport costs for patients) |
| **Facility/health system-level** | • On the job training, support supervision, clinical mentoring<br>• Functioning sample transportation channels in some countries | • Lack of trained staff<br>• Lack of space<br>• Burden of routine and/or study documentation |
| **Structural-level** | • *No structural-level facilitators reported by HCWs interviewed* | • Lack of awareness within the community<br>• Care behaviors (minor or grand-parents accompanying child)<br>• Cultural practices<br>• Stigma related to tuberculosis<br>• COVID-19-related disruption or interruption of services<br>• Transport and distance challenges<br>• Power cuts |

coverage of services among children in need. HCWs were enthusiastic about the nasopharyngeal aspiration procedure: though causing discomfort and pain to children, it was perceived as less invasive than other sample collection methods for children, as reported elsewhere [15,28,29]. Nasopharyngeal aspiration has been shown to be highly feasible in these decentralized contexts [30]. Most HCWs understood that even technical gestures could be performed at low levels of care and felt empowered to diagnose children. The importance of empowerment showed also in the fact that the acceptability of systematic screening seemed higher among HCWs who were responsible for further pursuing the diagnosis process, i.e. in PHCs of PHC-focused districts, and lesser when required to refer children for further tests, i.e. PHCs of DH-focused districts.

HCWs appreciated the innovative and technical aspect of nasopharyngeal aspiration, feeling privileged to be able to provide such level of care; together with the presence of Ultra in their facilities, nasopharyngeal aspiration was seen as contributing to promote their facility and the quality of services offered. Support supervision, clinical mentoring, and problem solving from experts were reported as contributing to HCWs confidence and self-efficacy in adopting these new diagnosis services, and overcoming initial challenges. Investments in supportive systems for HCWs will be key to the successful implementation of diagnosis of tuberculosis at decentralized levels of care [31].

Finally, HCWs shared overall positive perceptions and experience of microbiological testing, which seemed to be related to them being able to observe some children being diagnosed for tuberculosis based on Ultra results. Acceptability of an intervention is indeed often related to its direct tangible positive consequences: in an international survey conducted in 2017 among more than 700 clinical or laboratory experts, acceptance of novel diagnostic tests highly depended on test accuracy [32]. Yet, within the TB-Speed Decentralization study, microbiological yield was low [24].

At facility or health system level, acceptability of providing diagnosis services at low levels of health care was challenged by perceived workload. The burden of systematic screening due

to lack of sufficient HCWs and additional reporting and documentation tasks was reported almost unanimously among HCWs both at PHC and DH-levels. Shortage of staff in resource-limited settings is a common challenge and is indeed associated with limited time and capacity to deliver new services [33].

At a structural level, HCWs from PHCs shared frustration about not being able to collect stool samples or refer children to DH due to parents underestimating the need for care. Lack of community awareness about tuberculosis was reported as one of the factors related to delays in care seeking for tuberculosis and diagnosis in a study conducted in Indonesia [34]. Many HCWs further reported that stigma associated with tuberculosis and fear of COVID-19 may have prevented parents from accurately reporting on symptoms among their children at screening, as was shown recently in Ghana [35]. This was further challenged when children were accompanied by grandparents or siblings, who were not always well-informed about the child's health status and did not have parental authority to allow further TB diagnosis procedures. Transport, distance, and electricity supply were repeatedly mentioned as barriers to the acceptability of decentralization, hampering referral of patients–including for chest X-Ray, samples and test results between PHCs and DHs, in both directions. In this study distance from PHCs to DH ranged between 4.5 km and 86 km. Distance and transportation barriers are common challenges for tuberculosis programs in low- and middle-income countries [36,37]. Novel stool testing procedures with minimum processing steps [38] could be established at PHCs level and improve the acceptability, and feasibility, of decentralized childhood TB diagnosis. Overall across the 6 countries, distances from PHCs to DH ranged between 4.5 km to 86 km. Although it is likely that the decentralization of services at PHC-level reduced distance barriers for several patients, we were unable to formally compute the distance saved. Finally, limited and unstable electricity coverage further constrained the acceptability and feasibility of implementing chest X-Ray at DH. These findings highlight the complexity of the process of decentralization, that goes beyond the acceptability of services, and beyond the mere provision of services, with closer availability: decentralization of childhood tuberculosis diagnosis also touches on the capacity of a whole economic system to support this approach, and the capacity of end-users to take on the offer [39].

Our study has several limitations. First, our study sample was essentially pragmatic. We were not able to include in our study population representatives of all key HCWs involved, in each country, in decentralizing the comprehensive childhood tuberculosis diagnosis approach. This was mainly the case among medical doctors and skilled clinicians, unavailable for study participation due to high workload or being absent onsite during data collection. As a consequence, we may have missed some specific insight from key actors. Due to multi-country, multi-site, and multi-HCW profiles, we did not aim for data saturation in each specific setting and population. Second, we used a standardized interview guide, and SSRAs, although trained, may have been reluctant to explore in details some issues specific to their country. Third, country-specific expressions and views may have been simplified or distorted during the translation to English process. Fourth, the cultural differences between countries, in the way SSRAs conducted interviews and asked follow-up questions, and in the way HCWs expressed their ideas, feelings and experiences, may have influenced the interpretation of findings. And finally, the fact that we conducted a multi-country assessment of acceptability, highlighting common features across countries and not formally comparing countries, may have led to some over-simplification of findings, some being more relevant to certain settings than others and also maybe missing on interesting differences. These are common challenges reported during multi-country studies and analysis [40].

The multi-country design of our study also constitutes one of the major strengths of this work. Indeed, it provides insight into the perceptions and experiences of HCWs working in

different health care systems, different socioeconomic and cultural contexts, while representative of largely rural and poorly resources facilities in TB burden countries. Feasibility and acceptability of introducing innovative technologies in such districts is therefore representative of challenges that would be faced in most districts in project countries. Together with the large sample size, implementation of interventions in different sites and with different cadres of HCWs contributes to the generalizability of our study findings. Although this study was conducted in different COVID-19 contexts, i.e. during and after COVID-19 measures, we cannot rule out the fact that our findings may have been different in non-Covid times. For example, certain barriers and challenges such as transport challenges due to COVID-19 restrictions would not have been reported in non-COVID context and times. On the other hand, we may hypothesize that most perceptions and experiences among HCWs of the childhood TB diagnosis package, such as the feeling of being able to improving diagnosis, or the burden of perceived additional workload, would remain relevant regardless of COVID-19. Furthermore, we were able to interview HCWs, from both PHC and DH levels, who were directly involved in delivering the different components (from screening to clinical evaluation) of the diagnosis package, thus providing informed and first-hand feedback on their experience. Interviews were conducted by SSRAs who were not directly involved in implementing the study interventions, possibly facilitating free expression from HCWs, minimizing social desirability bias. Local SSRAs would have been able to recognize and better probe the country specific problems during interviews. We conducted multi-country analyses on the basis of first country-specific analyses and then overall study analysis, as a group of multi-country social scientists, combining, comparing and contrasting viewpoints and interpretations, thus maximizing the validity of findings. We used an existing and globally used framework of acceptability for analysis, thus contributing to the comparability of our findings with other studies. Finally, our study provides original data on the acceptability of decentralizing a comprehensive childhood tuberculosis diagnosis package from the perspective of frontline HCWs, and in a context of paucity of implementation data on how to implement childhood TB diagnosis approaches in the field.

## Conclusion

In this multi-country qualitative study, HCWs at DH and PHC-levels perceived and experienced decentralized childhood tuberculosis diagnosis as acceptable. Among others, they understood its purpose, it fit with their personal and professional values, and they were overall very motivated to provide what they perceived as improved quality of care. Acceptability however seemed to be hampered by several operational and feasibility issues, resulting in feelings of frustration or burden sometimes. Innovative referral mechanisms for patients, samples and chest X- could improve the acceptability of decentralized interventions for diagnosing childhood tuberculosis.

## Supporting information

**S1 File. Individual interview guide for Health care workers, <u>TB-Speed Decentralization study(2018–22).</u>**
(DOCX)

## Acknowledgments

The authors would like to thank all HCWs who took the time and consented to take part in the interview process, as well as all TB-Speed staff in the field who coordinated this qualitative study, many of whom are listed as part of the TB-Speed Decentralization Study Group. Special thanks to Monica Koroma from Solthis in Sierra Leone for her contribution to early stages of

the acceptability data collection. We thank the members of the TB-Speed Scientific Advisory Board who gave technical advice on the design of the study and approved the protocol (see TB-Speed Decentralization study group list); the Ministries of Health and national TB programs (NTPs) of participating countries; and the NTP district representatives who supported the TB-Speed Decentralization Study implementation.

## TB-Speed Decentralization Study Group

**Cambodia**

**Institut Pasteur du Cambodge, Epidemiology and Public Health Unit, Phnom Penh, Cambodia**: BORAND Laurence, DE LAUZANNE Agathe, DIM Bunnet, HEANG Seyla, KAING Sanary, KEANG Chanty, LY Socheat, MEAS Pichpiseth, NHOUENG Sovann, PRING Long, SRENG Vouchleang, YIN Song, SOVAN saren, PHAN Chanvirak, CHRENG Chanra, KHOUN Ratha, RIN Monicando, PAL Sophea, NANG Boraneath, POM Rathakrun

**National Center for Tuberculosis and Leprosy Control, Phnom Penh, Cambodia**: MAO Tan Eang

**District level decentralization**, **Batheay Operational District (OD)**

**Batheay Referral Hospital**

PI: CHHIM Simoy

Co-PI: TOUCH Huot, SUON Kosal, CHUM Saronn, TOK Kimhong, PRING Kimchorn, KROUCH Satya

Study nurses: CHOK Chean, SEUN Sunleng, PHON Savtey

Laboratory: NANG Mai, HUN Kimda, HONG Vanny, SOK Dara, CHEA Kosal

CXR team: CHHEANG Bunthoeun

**TUMNOB Health Center**

NOB Sorphea (head of health center), SEM Rino, LAY Lam

**PHAAV Health Center**

HAK Nareth (head of health center), SAY Haysan, KEM Pholly, MENG Sreyphal

**SAMBOR Health Center**

KONG Polak (head of health center), PHORN Sokheng, HIM Sreyvann, PHEACH Peakdey, KIVE Dalai

**CHHENG CHHNOK Health Center**

ROS Chamreun (head of health center), SAR Moeur, KONG Sreydy, KONG Seyha, YORN Sreytouch, TES Soam

**PHC level decentralization**, **Angrokar Operational District (OD)**

Angrokar Refefral Hospital

PI: KEP Sophal

Co-PIs: LENG Seroeung HENG Thy, NEAK Savorn, SENG Sim, PAY Pheakna, SUON Sithan, CHAN Sophanna

Study nurses: UM Dyna, SIN Savuth, PHAN Sam, KUM Sarim

Laboratory: KHATH Sokheng, PHEM Pong, SOK Seyha, NY Chanty

CXR team: LEIM Van, PICH Sereyvuth, CHHEANG Sengkry

**KUS Health Center**

SAY Nhoeun (head of health center), EANG Nhin, SAO Vannareth, SIM Vannak, SOM Sopheak, PONG Ney

**Nheng Nhang Health Center**

DOS Vichea (head of health center), VAN Sokha, SENG Sreyleap, YOEURNG Vanna, TOEM Kakada, KEO Thida

**TA PHEM Health Center**

KEO Sarom (head of health center), SEM Vuochny, VENG Sophal, RIN Chanthol, SEANG Vanny

**Tropang Anderk Health Center**

Koch Ny (head of health center), LOK Kiri, MAO Khemra, OUK Keovanna, MIN Maiya, MORM Suomun, KOY Rattana, CHHANN Sreypov, SET Sreytouch

**Cameroon**

**IRD, Yaoundé, Cameroon**:

AMBOUA SCHOUAME Audrey, BABEY Clifford, EDEN NGU Masama, GEUNOU Etienne, KWEDI NOLNA Sylvie, MBANG MASSOM Douglas, MELINGUI Bernard Fortune, NGA ELOMO Nadia, MVETUMBO Moïse, Nkembe Angeline

**Programme National de Lutte contre la Tuberculose (PNLT), Yaoundé, Cameroon**:

MBASSA Vincent, Ebo Krystel, Kuate Kuate Albert, Choupa Michelline, Mbede Maggi, Donkeng Valerie, Kamgaing Nelly

**Mother and child Centre, Chantal Biya Foundation, Yaoundé, Cameroon**:

TAGUEBUE Jean-Voisin

*PHC level decentralization*, *Obala operational District (OD)*

**Obala Hospital**

PI: Touha Yannick Achille

Co-PI: Abomo Zang Estelle

Study nurses: Bakoa Rosette, Mimbouombela Leger Esperance, Eleme Sabine, Eteme Marie Gabrielle, Elouna Nkoa Thierry, Ndzana Dieudonné

Laboratory: Bitti Christophe, Fotsing Olivier, Lani Boko Charlotte

CXR team: Mbonga Mathieu

*Primary Health Care of Obala health District*

**CSI Ngongo**

Atouga Fils Jean Baptiste (head of health center and Site PI), Njakou Sagang Ghislain Ulrich, Onomo Innocent, Essaga Hortense Charlotte

**CMA Batchenga**

Tetto Tanke Huvelin (head of health center and site PI), Mboudi Kouang Daniel Desiré, Eloundou Léon, Kengne Helene, Mbassi Felix

**CMC Fomakap**

Essama Sylviane Blanche Epse Ela (head of health center and Site PI), Bidjeme Juliette, Toua Eteme Hortense, Essama Nadege, Belinga Balla Roger

**CSI Essong**

Bikele Ntouda Carine (head of health center and site PI), Tassi Norbert, Nguiko Elsa Roline, Mekongo Leonard, EYEBE AYISSI Fabien

*DH level decentralization*, *Bafia Operational District (OD)*

**Bafia Hospital**

PI: BILOA Anaba Francine Christelle

Co-PIs: NSOM Philomène, YAM ESSOLA Celestin Géraud, MAME MOO Edwige Léa, MAKON Noé, Nounkep Yanghu Arllette Rita, Ebanga Frank, Assiga Ntsama Antoinette, Kamguia Djuimsop Carlyle Sorelle

Study nurses: MBABOU Diane, MAGUIP ABANDA Marie, Nguemafouo Doummene Rosine Berthe, Mekone Amos, Konfor Blessing Ngwankfu

**Laboratory**: MIMBOE

Jérome, TIONA Virginie, BELECK Roland

**CXR team**:

ZAM Sairou, ADIBONE Nicole

**TB Unit**:

Biaback Jean Claude

**Community health worker**:

Bessong Denis

Transport Motorbike: Aminou Gilbert

***Primary Health Care of Bafia Health District***

**CMA de Bokito**

Tchatat epse Kodoume Surya (head of health center and site PI), Bille Bonga Jeremie Pagnol, Fotso Monkap Aubin, HITEKELEK Epse NGON Annie, SEBE Vitrice, Makon Leo, Sebe Vitrice, Ennah Marie, Paul Boyolo Mpie

**CSI Messangssang**

Lwaboshi Kalumuna (head of health center), METCHOUM Diane Viviane (Site PI), Nzambe Celin, Dado Arnaud, Mbengang Milobert, Eyebi Marceline, Ngah Vanessa, Mballa Batonga Alice, Ayouba Solange, Ebode Pierrette, Mamou Majino

**CSI Balamba**

Tchoukogueu Deumaga Clarisse Flore Epse ABOUDE (head of health center and site PI), BOTOMOGNE BOMBA Marguerite, NGNET Salametou. Essengue Ngono Augustine Florence, Odionoloba Charles Rolland

**CMA Kiiki**

KAMGA José (head of health center and site PI), BISSO Bernice, Balemaken Ingrid Suzy, Mandoki a Bilong Marie Louise, Ndeng Ayouba Gertrude, NGON Josue

**Cote d'Ivoire**

**PACCI, Abidjan, Côte d'Ivoire**:

AKA BONY Roger, BAH Kacou Michel, BAKAYOKO Dro, BAKI Aimee Rolande, BANGA Marie-France Larissa, BOUZIÉ Olivier, BROU Kan, COULIBALY Pan, DANHO Serge, DELI Flavien, DION Alphonse, DO Bi, DOHOUN Armand, EDJEME William, Falé Cathérine, GOGOUA Saulé Melissa, KESSE Constant, KOMENA Auguste Eric, KOUADIO Christian, KOUAME Abel Arkason, MOH Raoul, NGUESSAN Marcelle Sandrine, SILOUÉ Bertine, Soua Nina, YAO Yapi Cyrille Prisca, OUASSA Timothée

**Programme National de Lutte contre la Tuberculose (PNLT), Abidjan, Côte d'Ivoire**: KOUAKOU Jacquemin

**France**

**University of Bordeaux, National Institute for Health and Medical Research (INSERM) UMR 1219, Research Institute for Sustainable Development (IRD) EMR 271, Bordeaux Population Health Centre, Bordeaux, France**:

BALESTRE Eric, BEUSCART Aurélie, CHARPIN Aurélie, D'ELBEE Marc, FONT Hélène, JOSHI Basant, KOSKAS Nicolas, MARCY Olivier, OCCELLI Estelle, ORNE-GLIEMANN Joanna, POUBLAN Julien, VERNOUX Elodie

**University of Montpellier, IRD, INSERM, TRANSVIH MI, Montpellier, France**: BONNET Maryline, CHAUVET Savine, LOUNNAS Manon

**Solthis, Paris, France**:

BRETON Guillaume

**TeAM SPI, France**:

NORVAL Pierre-Yves

**Mozambique**

**Instituto Nacional de Saúde, Maputo, Mozambique**:

CASSY Sheyla, CHAMBAL Verna, CHIÚLE Valter, CHIMBANJE Supinho, CUMBE Saniata, MATSINHE Mércia, KHOSA Celso, MABOTE Nairo, MACHAVA Salvador, MACHONISSE Emelva, MACUÁCUA Verónica, MILICE Denise, RIBEIRO Jorge, TIVANE Elcídio, UETELA Dorlim, VOSS DE LIMA Yara, ZANDAMELA Américo, ZITA Alcina

**Mozambique Ministry of Health, National Tuberculosis Control Program, Maputo, Mozambique**

MANHIÇA Ivan, JOSÉ Benedita

**José Macamo General Hospital, Maputo, Mozambique**

REGO Dalila

**University of California Los Angeles, David Geffen School of Medicine, Los Angeles, CA, USA**

BUCK W. Chris

**Chokwé District Hospital, Chókwe, Mozambique**

KASEMBE Kapoli, MASSANGAIE Atália, SITOE Assa, ARGOLA Ambostique, MIAMBO Césio, NHATSAVE Presequila

**Chalucuane Primary Health Center, Chókwe, Mozambique**

SITOE Gilda, VESTA Charifito

**Hókwe Primary Health Center, Chókwe, Mozambique**

DIMANDE Salvador, MAZEMBE, Lázaro

**Chiaquelane Pimary Health Center, Chókwe, Mozambique**

AMADE Nilza, CHAVELA Manuela, MACHEQUE, Nomsa

**Manjacaze District Hospital, Manjacaze, Mozambique**

COMÉ Salomão, MACHAVA Eulália, MUCAVELE Narciso, NHABANGA Jacinto, NICO-LAU Marlene, SIMBINE Natércia, UENDELA Lina

**Chidenguele Primary Health Center, Manjacaze, Mozambique**

JUAIO Micaela, SAÍDE Abiba

**Macuácua Primary Health Center Manjacaze, Mozambique**

MACIE, Naira

**Laranjeiras Primary Health Center, Manjacaze, Mozambique**

MONDLANE Fernando

**Chibonzane Primary Health Center, Manjacaze, Mozambique**

SIMANGO, Stélio

**Sierra Leone**

**Solthis, Freetown, Sierre Leone**:

BEYAN Prince, FLOMO Benjamin M., JALLOH Joseph Abubakarr, KAMARA Ishmael, KOROMA Monica G, LAMIN Mohamed, MATATA Lena, MUGISHA Jacob Ross, SENESIE Christiana M, SESAY Sheriff, TAMBA KAMARA Egerton

**Ola During Children's Hospital, Freetown, Sierra Leone**: MUSTAPHA Ayeshatu

**National Leprosy and TB Control Programme (NLTCP), Freetown, Sierra Leone**: FORAY Lynda

**Uganda**

**MUJHU Research Collaboration, MU-JHU Care Limited, Kampala, Uganda**:

AGONDEZE Sandra, KOBUSINGYE Agnes, NAMFUKA Mastula, NAMULINDA Faith, WOBUDEYA Eric

**Epicentre Mbarara Research Centre, Mbarara, Uganda**:

ARINAITWE Rinah, KAITANO Rodney, KASUJJA Martin, MWANGA-AMUMPAIRE Juliet, MWESIGWA Evans, NATUKUNDA Naome, NUWAMANYA Simpson, NYANGOMA Miria, ORIKIRIZA Patrick, TUMWIJUKYE Johnbosco, TURYASHEMERERWA Esther, Dan Nyehangane, Ivan Mugisha

**National Tuberculosis and Leprosy Program, Kampala, Uganda**: SEKADDE Moorine, TURYAHABWE Stavia

**Rakai District (DH focused)**

BIRYERI Winnie, Naika George, ONGWARA O Robert, NAJJUKO Allen, KAYIIRA Augustine, YAIRO Samuel, TUMWEBAZE Immaculate, NALWOGA Goreth

**Kanungu District (PHC focused)**

NSIYALETA Paul, AGABA Annet, MPIMBAZA M Martin, AKAMPURIRA Norbert, TUGUMISIRIZE Agatha, ARIYO Evans, AGABA Julius, NATUKUNDA Yovita, MUSAZI Nelson, MUSINGUZI Edmund, BALUKU Julius Brown

**Scientific Advisory Board**:

CHABALA Chishala (University of Zambia), CUEVAS Luis (Liverpool School of Tropical Medicine, UK), DELACOURT Christophe (Hôpital Necker-Enfants Malades, France), GRAHAM Steve (Chair; University of Melbourne, Melbourne, Australia), GRZEMSKA Malgorzata and VERKUIJL Sabine (WHO, Switzerland), HESSELING Anneke (Stellenbosch University, Cape Town, South Africa), MALECHE-OBIMBO Elizabeth (University of Nairobi, Kenya), MUSOKE Philippa (Makerere University, Uganda), NICOL Mark (University of Western Australia, Perth, Australia); MAO Tan Eang (CENAT, Phnom Penh, Cambodia).

## Author Contributions

**Conceptualization:** Basant Joshi, Laurence Borand, Jean-Voisin Taguebue, Raoul Moh, Celso Khosa, Guillaume Breton, Juliet Mwanga-Amumpaire, Maryline Bonnet, Eric Wobudeya, Olivier Marcy, Joanna Orne-Gliemann.

**Data curation:** Basant Joshi, Yara Voss De Lima, Douglas Mbang Massom, Sanary Kaing, Marie-France Banga, Joanna Orne-Gliemann.

**Formal analysis:** Basant Joshi, Yara Voss De Lima, Douglas Mbang Massom, Sanary Kaing, Marie-France Banga, Egerton Tamba Kamara, Sheriff Sesay, Joanna Orne-Gliemann.

**Investigation:** Basant Joshi, Maryline Bonnet, Eric Wobudeya, Olivier Marcy, Joanna Orne-Gliemann.

**Methodology:** Basant Joshi, Maryline Bonnet, Eric Wobudeya, Olivier Marcy, Joanna Orne-Gliemann.

**Project administration:** Basant Joshi, Joanna Orne-Gliemann.

**Resources:** Basant Joshi, Joanna Orne-Gliemann.

**Software:** Basant Joshi, Yara Voss De Lima, Douglas Mbang Massom, Sanary Kaing, Marie-France Banga, Egerton Tamba Kamara, Sheriff Sesay, Joanna Orne-Gliemann.

**Supervision:** Basant Joshi, Laurence Borand, Jean-Voisin Taguebue, Raoul Moh, Celso Khosa, Guillaume Breton, Juliet Mwanga-Amumpaire, Maryline Bonnet, Eric Wobudeya, Olivier Marcy, Joanna Orne-Gliemann.

**Validation:** Basant Joshi, Maryline Bonnet, Eric Wobudeya, Olivier Marcy, Joanna Orne-Gliemann.

**Visualization:** Basant Joshi, Maryline Bonnet, Eric Wobudeya, Olivier Marcy, Joanna Orne-Gliemann.

**Writing – original draft:** Basant Joshi.

**Writing – review & editing:** Basant Joshi, Yara Voss De Lima, Douglas Mbang Massom, Sanary Kaing, Marie-France Banga, Egerton Tamba Kamara, Sheriff Sesay, Laurence Borand, Jean-Voisin Taguebue, Raoul Moh, Celso Khosa, Guillaume Breton, Juliet Mwanga-Amumpaire, Maryline Bonnet, Eric Wobudeya, Olivier Marcy, Joanna Orne-Gliemann.

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
