## [Decision Letter · Decision Letter 0]

28 Feb 2023

PGPH-D-22-02101

Acceptability of decentralizing childhood tuberculosis diagnosis in low-income countries with high tuberculosis incidence: experiences and perceptions from health care workers in sub-Saharan Africa and South-East Asia

Dear Dr. Joshi,

Thank you for submitting your manuscript to PLOS Global Public Health. After careful consideration, we feel that it has merit but does not fully meet PLOS Global Public Health’s publication criteria as it currently stands. Therefore, we invite you to submit a revised version of the manuscript that addresses the points raised during the review process.

In your revision, please address all reviewer comments in a point-by-point fashion, paying particular attention to the points of Reviewer #2 which address context and interpretation of your results.

We look forward to receiving your revised manuscript.

Kind regards,

Helen Dimaras, PhD

Academic Editor

Journal Requirements:

1. Please send a completed 'Competing Interests' statement, including any COIs declared by your co-authors. If you have no competing interests to declare, please state "The authors have declared that no competing interests exist". Otherwise please declare all competing interests beginning with the statement "I have read the journal's policy and the authors of this manuscript have the following competing interests:"

b. If any authors received a salary from any of your funders, please state which authors and which funders.

Additional Editor Comments (if provided):

Reviewers' comments:

Reviewer's Responses to Questions

**Comments to the Author**

1. Does this manuscript meet PLOS Global Public Health’s publication criteria? Is the manuscript technically sound, and do the data support the conclusions? The manuscript must describe methodologically and ethically rigorous research with conclusions that are appropriately drawn based on the data presented.

Reviewer #1: Yes

Reviewer #2: Yes

2. Has the statistical analysis been performed appropriately and rigorously?

Reviewer #1: N/A

Reviewer #2: N/A

3. Have the authors made all data underlying the findings in their manuscript fully available (please refer to the Data Availability Statement at the start of the manuscript PDF file)?

Reviewer #1: Yes

Reviewer #2: No

4. Is the manuscript presented in an intelligible fashion and written in standard English?

Reviewer #1: Yes

Reviewer #2: Yes

5. Review Comments to the Author

Reviewer #1: This is an interesting, thorough, and relevant paper. I just have a few comments and questions:

1. This study was carried out during the acute phase of the COVID-19 pandemic when TB testing in many places was down because people were reluctant to bring their child to a healthcare facility and some healthcare workers were reluctant to obtain sputum. Although the pandemic is mentioned a couple of times, it would be useful to include something in the discussion acknowledging the ways in which the conclusions from this paper might or might not be relevant going forward as things return to normal.

2. Were there any differences in results by country?

3. Given that some patients were referred from the PHC to the DH, it would be good to provide some context around how far these services were from each other? How logistically difficult and time-consuming was it for parents/caregivers to do this?

4. On a similar note, part of the point of decentralizing services is so that patients do not have to travel as far – in these settings, what were these distance savings like? How far did they have to go for centralized services as compared with decentralized services?

5. How do these results generalize to settings where resources like digital chest x-ray do not exist at the decentralized service facilities? Are the PHCs and DHs included in this study comparable to some more rural and poorly resourced settings in high TB burden countries?

Reviewer #2: General Comments

In this large, qualitative study, the investigators conducted key informant interviews with 130 healthcare workers participating in the TB-Speed Decentralization study, a pre-post implementation study of a multi-component pediatric TB diagnostic package in 6 countries in sub-Saharan Africa and Southeast Asia. The goals of this sub-study were to characterize the acceptability to healthcare workers of decentralizing TB diagnostic evaluation to primary health centres and district hospitals, including introducing systematic TB screening, collection of nasopharyngeal aspirates and stool specimens and molecular testing with GeneXpert MTB/RIF Ultra, as well as enhanced clinical evaluation, sometimes with radiographic evaluation and referral to district hospitals. After carrying out thematic analysis and deductive coding using the Theoretical Framework of Acceptability, the authors show that the package was broadly acceptable.

Major Comments

This is a well-conceived, and generally well-executed study that offers valuable context on health worker perceptions of a broad package of pediatric TB diagnostics implemented in real-world settings in multiple high-burden countries. I have a few questions for the authors on the reporting and interpretation of the study

1. It would be useful to have more information on the setting and context of the parent study, in particular a) what was the general pre-intervention standard of care for pediatric TB evaluation at these sites, and b) how were the different decentralization packages implemented in the different countries. Table 1 is helpful on the different intervention components, but it’s not quite clear where or why different models (i.e. PHC-focused versus DH-focused) are being used, and who is providing which services where. This need not be a comprehensive description; in fact, just a short overarching explanation so readers don’t necessarily have to study Table 1 to understand the intervention. Is it accurate to say that in the DH-focused strategy, only symptom screening is decentralized, while in the PHC-focused strategy, screening and diagnosis are decentralized and DH referral is only necessary for chest radiography?

2. Building on #1, the Results subheadings summarize the key components of the intervention and effectively link the acceptability findings to the intervention components. In addition to better describing who is doing these steps, would suggest using parallel terminology and ordering of the Table 1 and Results headings (currently they are only loosely aligned).

3. In the Methods, under study population, could you explain which axes of diversity guided your purposive sampling? Gender is mentioned, but it is not clear if health workers were sampled by profession or by the role that they played in offering pediatric TB services? If the later, it would help to describe this distribution in Table 2. Could you also comment in the Methods or Results as appropriate on the distribution of health worker professions by country (e.g., in Cambodia and Uganda, a much broader cross-section of healthcare workers participated). This would better set the stage for understanding the limitation you mention that doctors and other clinicians were not included in some countries even though it seems that you had planned to interview them.

4. Appendix 1 shows and interview guide with 36 questions, a very large number for a single sitting. Was the instrument piloted, how were the interviewers trained to ask the questions, and what was the duration of the interviews? Were the interviews completed in a single session?

5. Did the authors assess for thematic saturation?

6. There is only loose alignment between the findings in Table 4 and the summary of the multi-level factors in the Discussion session. For example, under the individual factors, I don't find the particular notion of ethics discussed in the Results or the Tables (where ethics is discussed in terms of patient acceptance and discomfort). Moreover, there is a discussion about PHC-focused HCW being more engaged that DH-focused strata, but most of if not all of the quotes in the Results and in Table 3 show a high level of engagement in DH-focused facilities. In the Discussion of facility or health system level factors, the factors that were acceptable (or facilitated acceptability) are not mentioned.

Aligning these would help ensure the Discussion / Conclusions are linked to the primary findings presented in this manuscript (or in the literature), and it would be clearer if the key findings of the Discussion were more closely aligned to and ordered after the themes addressed in Table 4.

Minor Comments:

1. Line 55: Consider specifying "stool collection" to make the reference to “cultural practices” a little clearer

2. Line 337: Could you explain the meaning of the word collaborators? Who are they describing? Is the speaker suggesting that these individuals accompany families all the way from the PHC to the DH?

3. Line 424: Suggest changing "was shown" (which implies "in this study") to "has been shown in other contexts", or similar.

4. Line 480: Could you elaborate on why you think SSRAs may not have uncovered all the relevant themes? One might assume the opposite, if SSRAs from the same country are better able to recognize and probe on local nuances.

5. Line 481, in the same vein as the previous comments, a diversity of sites and perspectives would not be a weakness, but a strength from a qualitative methods perspective.

6. Table 1: “Greater than”, or “greater than or equal to” 2 weeks? “Documented” or “Reported” weight loss? D7=Day 7? After presentation? After treatment initiation?

7. Line 509: “every sense of the word” might be better stated – the authors found that the strategy was broadly acceptable across all 7 domains of the Theoretical Framework of Acceptability, but also (as said at the opening of the Discussion) elements that were not acceptable (or were barriers to acceptability). Similarly, the statements about sustainability or scalability seem out of scope in that they imply that strategies to promote acceptability would also enhance sustainability or scalability.

6. PLOS authors have the option to publish the peer review history of their article (what does this mean?). If published, this will include your full peer review and any attached files.

**Do you want your identity to be public for this peer review?** For information about this choice, including consent withdrawal, please see our Privacy Policy.

Reviewer #1: No

Reviewer #2: No

---

## [Editor Report · Decision Letter 1]

6 Jun 2023

PGPH-D-22-02101R1

Acceptability of decentralizing childhood tuberculosis diagnosis in low-income countries with high tuberculosis incidence: experiences and perceptions from health care workers in sub-Saharan Africa and South-East Asia

Dear Dr. Joshi,

Thank you for submitting your manuscript to PLOS Global Public Health. After careful consideration, we feel that it has merit but does not fully meet PLOS Global Public Health’s publication criteria as it currently stands. Therefore, we invite you to submit a revised version of the manuscript that addresses the points raised during the review process.

It was noted that some comments were addressed in the point by point response, but skipped or addressed with insufficient detail in the revised manuscript. Please see comments below.

We look forward to receiving your revised manuscript.

Kind regards,

Helen Dimaras, PhD

Academic Editor

Journal Requirements:

2. We have noticed that you have uploaded Supporting Information files, but you have not included a list of legends. Please add a full list of legends for your Supporting Information files after the references list. 

Additional Editor Comments (if provided):

Thank you for providing a revised manuscript with point by point response. It was noted that some comments were addressed in the point by point response, but skipped or addressed with insufficient detail in the revised manuscript.

Please revise the manuscript to address the following reviewer comments in detail:

-Reviewer 1, comment 1: address specific ways in which conclusions may or may not be relevant to post-covid times

-Reviewer 1, comment 4

-Reviewer 1, comment 5: address with further detail in manuscript

-Reviewer 2, comment 1b

-Reviewer 2, comment 4

-Reviewer 2, comment 5: the authors say they can't "ensure saturation was reached", which is an odd statement. Typically interviews are stopped when the interviewers hear the same comments repeatedly. Why did the authors stop the interviews prior to this? Or do the authors simply mean that saturation was reached from the participants in question, but further insight may have been aqcuired had a more diverse group been recruited?
---

## [Editor Report · Decision Letter 2]

6 Sep 2023

Acceptability of decentralizing childhood tuberculosis diagnosis in low-income countries with high tuberculosis incidence: experiences and perceptions from health care workers in sub-Saharan Africa and South-East Asia

PGPH-D-22-02101R2

Dear Mr Joshi,

We are pleased to inform you that your manuscript 'Acceptability of decentralizing childhood tuberculosis diagnosis in low-income countries with high tuberculosis incidence: experiences and perceptions from health care workers in sub-Saharan Africa and South-East Asia' has been provisionally accepted for publication in PLOS Global Public Health.

Best regards,

Julia Robinson

Executive Editor